# Processing of positive newborn screening results: a qualitative exploration of current practice in England

Jane Chudleigh ![ORCID],[1] Holly Chinnery,[2] Pru Holder,[1] Rachel S Carling,[3] Kevin Southern,[4] Ellinor Olander,[1] Louise Moody,[5] Stephen Morris ![ORCID],[6] Fiona Ulph,[7] Mandy Bryon,[8] Alan Simpson[9]

For numbered affiliations see end of article.

**Correspondence to**
Dr Jane Chudleigh;
j.chudleigh@city.ac.uk

## ABSTRACT

**Objective** To explore current communication practices for positive newborn screening results from the newborn bloodspot screening (NBS) laboratory to clinicians to highlight differences, understand how the pathways are implemented in practice, identify barriers and facilitators and make recommendations for future practice and research.

**Design** A qualitative exploratory design was employed using semi-structured interviews.

**Setting** Thirteen NBS laboratories in England.

**Participants** Seventy-one clinicians; 22 NBS laboratory staff across 13 laboratories and 49 members of relevant clinical teams were interviewed.

**Results** Assurance of quality and consistency was a priority for all NBS laboratories. Findings indicated variation in approaches to communicating positive NBS results from laboratories to clinical teams. This was particularly evident for congenital hypothyroidism and was largely influenced by local arrangements, resources and the fact individual laboratories had detailed standard operating procedures for how they work. Obtaining feedback from clinical teams to the laboratory after the child had been seen could be challenging and time-consuming for those involved. Pathways for communicating carrier results for cystic fibrosis and sickle cell disease could be ambiguous and inconsistent which in turn could hamper the laboratories efforts to obtain timely feedback regarding whether or not the result had been communicated to the family. Communication pathways for positive NBS results between laboratories and clinical teams could therefore be time-consuming and resource-intensive.

**Conclusion** The importance placed on ensuring positive NBS results were communicated effectively and in a timely fashion from the laboratory to the clinical team was evident from all participants. However, variation existed in terms of the processes used to report positive NBS results to clinical teams and the people involved. Variant practice identified may reflect local needs, but more often reflected local resources and a more consistent 'best practice' approach is required, not just in the UK but perhaps globally.

**Trial registration number** ISRCTN15330120.

## Strengths and limitations of this study

► This is the first known study that has explored communication pathways for positive newborn bloodspot screening (NBS) results from the laboratory to clinical teams.
► Participants represented the 13 NBS laboratories in England involved in managing the nine conditions currently included in the NBS programme increasing the transferability of the findings.
► The study design, data collection and analysis were influenced by members of the patient and public involvement advisory group and relevant charities.
► The researchers are experienced in this field which may have biassed data collection and analysis.

## INTRODUCTION

In the UK, newborn bloodspot screening (NBS) comprises a heel prick blood sample taken via a NBS card between days 5 to 8 of life to identify pre-symptomatic babies that are affected by genetic or congenital conditions.[1] It is known that early diagnosis leads to better health outcomes for the child.[1 2] In the UK, since January 2015, NBS covers nine conditions (table 1).

Each year, around 10 000 babies born in the UK have a positive NBS result. These positive NBS results include both babies who will eventually be diagnosed as being affected by one of the nine life-changing conditions currently screened for[3 4] (n=approximately 1500), those who will later be confirmed as gene carriers for sickle cell disorder (SCD) or cystic fibrosis (CF) but unaffected by the disease and babies who screen positive but for whom screened disease is ruled out (false-positive results).

### Communicating presumptive positive NBS results

NBS laboratory guides are available for each of the conditions included in the NBS

**Table 1** Conditions included in the newborn bloodspot screening programme

| Name of condition | Abbreviation |
| --- | --- |
| Sickle cell disorder | SCD |
| Cystic fibrosis | CF |
| Phenylketonuria | PKU |
| Homocystinuria | HCU |
| Glutaric aciduria type 1 | GA1 |
| Medium chain acyl CoA dehydrogenase deficiency | MCADD |
| Maple syrup urine disease | MSUD |
| Isovaleric aciduria | IVA |
| Congenital hypothyroidism | CHT |

Programme in the UK.[5–9] Positive NBS results are referred to as being 'presumptive positive' (PP) results. That is, the result is presumed to indicate a positive result until further diagnostic testing is undertaken to either confirm or rule out the condition being screened for. Within each condition, there is specific information on how PP cases should be reported and communicated. While some aspects of the pathways are similar, some are very different such as the time frame for the first clinic appointment. Table 2 summarises the different pathways as established in the guides, for each of the screened conditions.

Communicating positive NBS results is not an event but a process that starts from the moment the result is identified by the NBS laboratory as being above the agreed analytical 'cut-off' and ends when the parents are given the definitive diagnosis for their child.[10] Most literature to date has focussed on how positive NBS results are communicated to families via appropriate clinicians.[11–16] These studies have highlighted that despite guidance, this occurs in a range of ways which are not currently well-defined. In addition, poor, or inappropriate, strategies for communication of positive NBS result to families can influence parental outcomes in the short term[11 13 16–19] but may also have a longer term impact on children and families.[20]

The NBS laboratory guides[5–9] provide detailed explanations regarding the analytical aspects of NBS, quality assurance, clinical referral and follow-up and reporting and data collection for each of the screened conditions. However, no literature has explored how the process of communication from the laboratory to clinical teams is implemented in practice and how this might influence communication with families.

The purpose of the current study was to explore current communication practices for positive newborn screening results from the NBS laboratory to clinicians to highlight differences, understand how the pathways are implemented in practice, identify barriers and facilitators and make recommendations for future practice. Health professionals' experiences of communicating positive NBS results to families have been reported elsewhere.[10]

## METHODS

A qualitative exploratory design was utilised using semi-structured telephone interviews with laboratory staff employed in the 13 NBS laboratories in England and members of relevant clinical teams notified of positive NBS results from the respective NBS laboratories. This study is part of an ongoing programme of work[21] and was approved by the London Stanmore ethics committee (17/LO/2102).

### Patient and public involvement

Patient and public involvement (PPI) was instrumental in the design and conduct of this study. Eight parents of babies who had received a positive NBS screening result for one of the nine screened conditions formed a PPI advisory group who met prior to, during and following data collection. Their suggestions were incorporated into the study design, the data collection tools and the data analysis and presentation. The PPI group were presented with data from the annual reports of the NBS Programmes and made suggestions as to which sites should be used in later phases of the study design. The PPI group also suggested clarifying that positive NBS results indicated an 'abnormal' result while negative NBS results indicated a 'normal' result during data collection, as they had found this confusing when they received their child's NBS result. Initial findings of were presented to member of the PPI group during regular 6-month meetings and drafts of manuscripts were also shared with PPI members to ensure these were presented in a readable format. In addition, we obtained the views of representatives from charities for the screened conditions including Metabolic Support UK, the British Thyroid Foundation, the CF Trust and the Sickle Cell Society.

### Setting

In England, there are 13 Newborn Screening laboratories that process the results for the nine conditions that are currently included in the NBS Programmes, these comprised the study sites.

### Inclusion and exclusion criteria

Staff employed in NBS laboratories and involved in the processing of positive NBS results and members of relevant clinical teams who had received a positive NBS from NBS laboratories in the previous 6 months were included. Staff who had not been involved in processing or receiving positive NBS results in the last 6 months or who had personal experience of receiving a positive NBS result were excluded.

### Recruitment and sampling

A two-stage sampling approach was employed where participants were first sampled purposively based on their experience with the phenomena of interest, followed by a second stage of snowball sampling where the first participants suggested others. Directors of all 13 NBS laboratories in England were invited to participate. These were identified through the UK Newborn Screening

**Table 2** UK guidelines for communicating positive NBS results from laboratories to clinical trams following a PP NBS result

| Condition(s) | Method of referral from laboratory to clinical team | Time frame for communicating a PP result to the clinical team | Requirements when communicating PP results to families | Time frame for first clinic appointment |
|---|---|---|---|---|
| Congenital hypothyroidism[8] | Verbally and in writing by secure email, including a link to the standardised diagnostic and initial treatment protocol, to either a paediatric endocrine team or a lead paediatrician with a special interest in CHT | Same or next working day of the definitive NBS result being available | Communicated to the family by an 'informed health professional' who should provide the family with the appropriate information leaflet available via the screening programme and the child's appointment details | Must take place on the same day or the next day after parents are informed of their baby's positive NBS result |
| Cystic fibrosis[5 9] | Refer the baby immediately to a designated person at the appropriate Regional CF Centre. The regional CF centre will usually be responsible for liaising with local CF clinics | Within one working day of the definitive screening result | Parents should be informed by an 'appropriate health professional' but not on a Friday, Saturday or Sunday and that ideally Thursday should be avoided as well. The family should be contacted via a structured telephone call in the first instance informing them that the result will be discussed later that day by a healthcare professional with appropriate experience and support to give bad news. Following this, the guidance states that the positive CF screening result may be given at a home visit but sometimes a second structured phone call might be used. In both instances, it is recommended this should be done by a healthcare professional with knowledge of screening and CF | Parents should be offered an appointment on the following day after they are informed of the PP result and the diagnostic assessment within five working days of receiving the result |
| Sickle cell disease[7] | Must be reported as per local pathways, to the clinician and/or designated sickle cell and thalassaemia centre | Not specified | Local protocols must ensure that PP results are given to parents by a trained healthcare professional face-to-face by 28 days of age | The baby should enter the care of a specialist haemoglobinopathy centre by 90 days of age |

Continued

**Table 2**  Continued

| Condition(s) | Method of referral from laboratory to clinical team | Time frame for communicating a PP result to the clinical team | Requirements when communicating PP results to families | Time frame for first clinic appointment |
|---|---|---|---|---|
| Inherited metabolic diseases (IMDs) including (MCADD, MSUD, PKU, IVA, HCU and GA1)[6] | Referral of all IMD PP patients should be undertaken in co-ordination with a clinical service that complies with the service specification for a paediatric IMD centre to ensure continuity of specialist care; PP cases should be reported to the specialist clinical team (or designated local team for PKU) verbally and in writing | Same working day that the PP result is available | Parents should be informed of the PP NBS result face-to-face, usually at home by a member of the relevant IMD team and provided with the relevant 'suspected leaflet' and details of the first clinic appointment (PKU). Provided with the relevant 'suspected leaflet' and details of the first clinic appointment via email (MCADD), by a member of the IMD specialist team (HCU) who should also liaise with the appropriate hospital until the baby can be transferred to the specialist IMD centre (MSUD, IVA and GA1) | The baby should be: reviewed at an appropriate hospital within 24 hours (MCADD) or on the next working day via web link or in person (HCU); or the same or next working day (PKU); or taken to an appropriate hospital (if not an inpatient already) where initial assessment will take place and then transferred to a specialist IMD centre on the same day the PP screening result is available. If transfer is not available, the specialist team will liaise with the local hospital to arrange diagnostic testing and supply of dietary supplements (MSUD, IVA and GA1) |

CF, cystic fibrosis; CHT, congenital hypothyroidism ; GA1, glutaric aciduria type 1; HCU, homocystinuria; IMD, inherited metabolic diseases; IVA, isovaleric aciduria; MCADD, medium chain acyl CoA dehydrogenase deficiency; MSUD, maple syrup urine disease; NBS, newborn bloodspot screening; PKU, phenylketonuria; PP, presumptive positive.

Laboratories Network (http://www.newbornscreening.org/site/laboratory-directory.asp) and were contacted via email by a member of the research team. Directors of newborn screening laboratories were invited to be the local principal investigator for their study site and were asked to provide names and contact details of staff within the laboratory who met the inclusion criteria for the study. These staff members were contacted via email and invited to participate.

Representative members of local clinical teams (medical consultants; general paediatricians; nurse specialists; health visitors; specialist screening nurses and genetic counsellors) were identified through individual trust websites and contacted via email to be invited to participate. Those who agreed to participate were also asked to identify other members of their team who they thought should be interviewed to provide further information about the NBS process. These additional potential participants were also contacted by email and invited to participate. Written informed consent was obtained from all participants.

### Data collection

Semi-structured telephone interviews comprising closed and open-ended questions were conducted by JC between June 2018 and February 2019 to identify the approaches used to communicate positive NBS results from NBS laboratories to health professionals (online supplemental file 1). Data were collected on: the mode of communication strategy (face-to-face; letter; telephone; and e-mail); the resources involved in each communication strategy; who provides the information and their role; and location (co-located or alternative site) of relevant services for each condition.

### Data analysis

The purpose of data analysis was to describe and identify approaches currently used to communicate positive NBS results from NBS laboratories to health professionals. Qualitative data from open-ended questions were analysed using thematic analysis[22] with an inductive approach. Data from laboratory staff and clinical staff were analysed separately. Seven interview transcripts from laboratory staff were coded by two members of the research team (JC and HC) to aid in coding comparisons and inform and align code development.[23] A code book was developed based on these jointly coded transcripts. A further seven laboratory transcripts were then coded separately by the same two members of the research team using the code book. These separately coded transcripts were compared, a similar process was followed for the transcripts for clinical staff. Following this, the same two members of the research team coded the remainder of the laboratory and clinical staff transcripts using the relevant code books. This was an ongoing, iterative process; new codes were developed and the definition of codes refined as analysis progressed.[24] Once this initial coding had been completed, these codes were then collapsed into themes.

**Table 3** Demographics of participants

**NBS laboratory staff**

| Profession | | Number of staff interviewed |
|---|---|---|
| Director of NBS laboratory | | 9 |
| Consultant biochemist/haematologist | | 2 |
| Senior/ clinical scientist | | 11 |
| Length of service | Median 12.0 years | Range 1.0 to 22.0 years |
| Length of interview | Median 30.8 min | Range 13.3 to 45.1 mins |

Clinical teams

| Profession | | Number of staff interviewed |
|---|---|---|
| Medical consultant | | 21 |
| Clinical nurse specialist | | 21 |
| Screening specialist nurse/midwife | | 5 |
| Service coordinator | | 1 |
| Paediatric dietician | | 1 |
| Length of service | Median 11.0 years | Range 1.5 to 23.0 years |
| Length of interview | Median 33.4 min | Range 10.4 to 54.6 mins |

NBS, newborn blodspot screening.

Communication pathways for each condition for each of the 13 NBS laboratories (unit of analysis) were also developed (online supplemental file 2).

### Positionality and reflexivity

Members of the study team (JC, LM, FU, MB and KWS) have been involved in or continue to undertake a variety of roles and activities associated with the NBS Programme in the UK. It is acknowledged that this could have led to potential bias during data collection and analysis. However, this was balanced by other members of the research team who had previously had minimal involvement in NBS (HC, EKO, AS, SM and PH). Data collection was undertaken by JC and data analysis was mainly undertaken by JC and HC who fall within both camps. Neither JC nor HC were employed in the organisations where data collection was undertaken.

## RESULTS

In total, 71 interviews were conducted: 22 with NBS laboratory staff across 13 laboratories and 49 with members of clinical teams. Four eligible participants declined to be interviewed. Demographics of participants can be seen in table 3.

### Themes

Five themes were identified from the data: the importance of the result to the child and family; different referral

approaches for different conditions; no unified process to provide feedback to laboratories; providing carrier results; inconsistency and ambiguity and resource use and responsibilities. These are explored in detail below and supported by illustrative quotations from interview data.

### The importance of the result to the child and family

Assurance of quality and consistency was a clear priority for all NBS laboratories. This applied to all aspects of analysis, including quality checks of the NBS card, data entry, first line and, if required, second line testing, and timeliness of reporting. There was also a clear appreciation of the significance of the results for families.

> …everybody is aware there is a baby at the end of this, there's a family at the end of this and everybody wants to do the best by that family. So, I think because of that everybody pulls together really well and makes sure, even if it's bad timing, which it usually is, everybody still pulls together to make sure it happens. Site 13

In addition, laboratory staff clearly appreciated the urgency of communicating positive NBS results to clinicians particularly for the inherited metabolic diseases (IMDs), including medium chain acyl CoA dehydrogenase deficiency (MCADD), maple syrup urine disease (MSUD), phenylketonuria (PKU), isovaleric aciduria (IVA), homocystinuria (HCU) and glutaric aciduria type 1 (GA1).

> Any of the metabolic conditions, if we see a raised result for first time, we will repeat that immediately. So, rather than it going through the system and being repeated, and us only getting the second result the next day, we'd repeat it immediately, so we repeat it effectively offline, and then if it turns out to be positive, we would make the referral straight away. Site 8.

### Different referral approaches for different conditions

Even though national guidelines are available from Public Health England for referring positive NBS results from the laboratory to the relevant clinicians, 10 out of 13 of the laboratories created their own templates for this purpose following further development and improvement by staff:

> … we have a proforma that we send …when we get the results. I know there's a national one which we've looked at but we think our own one probably ticks more boxes. From our point of view, it seems to work for us better Site 7

When a positive NBS result occurs, referrals are made to a range of different clinicians including condition specific consultants, their secretaries, condition specific specialist nurses, specialist screening nurses or screening co-ordinators. The referral process was often based on local arrangements, resources and the fact individual

laboratories had detailed standard operating procedures for how they work. These could be quite complex with variation for each condition and often the need to cover large geographical areas. However, usually, these were well understood and implemented.

> …we've got SOPs (standard operating procedures) that define where we need to call for whichever result in whichever place…for our congenital hypothyroidism, we have named consultants at (Hospital E). We have a name for (Hospital F) and (Hospitals G, H, I and J) where we'll just report to an on-call paediatrician, So, PKU only if they're (Hospital J) babies we report to a (Hospital J) paediatrician, or there's one of two (Hospital J) paediatricians. If we can't get them, then it goes to the metabolic team at the (Hospital A). Then, MCADD babies if they are (Hospital E) or (Hospital J), then we can report them either to a (Hospital E) named clinician or the (Hospital J) named clinicians. Again, if they're not available, then they'll go down to the metabolic team at (Hospital A) Site 3

Many factors were seen to promote effective communication between the laboratory and clinical teams. This included teams being small which meant everyone within the team knew one another and understood each other's roles; the proximity of teams within sites made communication easier between laboratory staff and relevant clinical teams; and good working relationships among teams within the same hospital and across sites.

> I think the close relationship between all the different professional groups for this part of the country works extremely well. Everybody tends to know each other personally; it means there are no barriers to discussion… Now, increasingly, we're working as closely as we can with (Hospital D) in particular…they'll all cover each other, and the labs work quite closely together. We have joint meetings occasionally and to some extent, we have a joint strategic vision because there are specialist lab things that each one of the three of us do. So, we're trying to develop separate areas and compliment rather than compete. Site 9

However, for many laboratories, referral of positive NBS results for congenital hypothyroidism (CHT) was viewed as more problematic. For all of the other screened conditions, dedicated condition specific specialist clinical teams are available to receive the positive NBS result. However, while some babies who have a positive NBS result for CHT will be seen within specialist endocrine teams, CHT is generally viewed as being possible to manage by general paediatricians and, therefore, often babies will be referred to local paediatric services. This can mean that in some instances, there is not a named individual to act as a point of contact.

> I'm having to go through a switchboard at a different hospital to try and find somebody who might not be

in that hospital because their clinic's in another hospital… I would love it if I just had one person to call about all my hypothyroidism babies, make my life so much easier if I didn't have to phone different GPs and different consultant endocrinologists. Site 10

Despite these challenges, laboratory staff were acutely aware of the importance of obtaining this feedback to ensure the screening process was complete. Therefore, even though it meant that they often had to invest additional time, laboratory staff would ensure that feedback was received so the final outcome (true positive, false positive and carrier) could be recorded accurately and in a timely manner. Concerns were also raised by laboratory staff and members of clinical teams about the equity of care particularly in relation to availability of scans following a positive NBS result for CHT.

So in terms of equity of care, it would seem that, given that it's a national screening programme, people should be having the same tests for diagnosis as well. Site 4

Conversely, communication of positive CHT NBS results to families was seen as being more straightforward to manage than some of the other conditions included in the NBS programme.

We do far less visits for CHT babies. They're mainly phoned up and told about the results, and then told when the appointment is and where …It's a lot simpler disorder. Site 1

### No unified process to provide feedback to laboratories

In the UK, following the referral of a baby with a positive screening result, NBS laboratories require feedback from the relevant clinical team once the baby has been seen, assessed and confirmatory testing has been undertaken. There was no consistent unified national approach to providing this feedback, which led to time-consuming efforts by the laboratory staff to obtain information. In contrast, the responsible clinicians often felt they had a clear idea of their duties in this regard.

…if we don't receive feedback, we have to phone and write letter, and that does take quite a bit of time. Site 1

This was in contrast to the views of clinicians who were responsible for providing the feedback to the NBS laboratories who described steps they took to ensure this information was fed back to the laboratories. This suggests that there may be a mismatch between the information the laboratories actually require, the information clinicians are providing and who is seen to have ownership of the information.

Then, what I will normally do then is email (the NBS laboratory) back to say, 'Yes, the parents will be attending,' and so on, or, if the parents declined, which

has never happened, 'Okay, they're not coming,' and I would assume they would follow-up. Site 7

The ability of the laboratory to collate and coordinate feedback from different sources, after a child had been seen, was considered to be particularly challenging and time-consuming for CHT. This was often attributed to the fact that affected babies were often seen in 'local' centres rather than tertiary referral centres due to the treatable nature of the condition. As a result, clinicians from many more localities might be involved in their care. To remedy this, some laboratories had sought local solutions to help them deal with the difficulties associated with feedback for positive NBS results for CHT.

CHT is much more of a problem in this region because we are not phoning one individual consultant in this region…The CHT is, I have to chase around a lot more to get that information from other hospitals. They are not very forthcoming, often, at providing the information so I have to chase around, and often I have to get my consultant colleague here, the endocrinologist, to help me with that because that can be a real challenge. Site 10

This was in contrast to the IMDs where the conditions were viewed as being more urgent due to the potential life-threatening nature of, particularly, MCADD, MSUD and IVA. Also, because affected babies were only seen at tertiary centres where often the NBS laboratory staff worked closely both physically and personally with IMD clinical team members.

…for the IMD conditions, because the consultant that sees them is based in the same hospital as me (Laboratory Director) and the fact he is my colleague, we work together to provide the IMD clinical service, I do get that information from him. I also can see that the diagnostic blood tests have been received in our laboratory. So, I know those babies have been seen. Site 10

### Providing carrier results; inconsistency and ambiguity

Pathways for communicating carrier results to families were viewed by some laboratories as ambiguous and inconsistent. NBS laboratories expressed concern regarding whether parents had been informed of their baby's SCD or CF carrier status and by whom, as this information was often difficult for them to ascertain. Carriers are healthy children who carry a single faulty gene. The most commonly received positive newborn screening result is for carrier status for SCD. Communication pathways can include a range of health professionals, including sickle cell coordinators, sickle cell counsellors, general practitioners, and/or health visitors via telephone, letter, home visit or during the baby's health review when they are 6 weeks of age. Therefore, although this represents a different result scenario (the baby is considered healthy),

there is still evidence of inconsistency in terms of the pathways used.

> … for a child who has who has been found to have a carrier status of sickle cell, we just correspond with the health visitor and the GP in a non-urgent way because that is something that just needs to be added to their record but probably won't impact on their health greatly. Site 2

The purpose of the CF NBS protocol in the UK is to maximise the detection of affected individuals (those with two disease causing mutations of the CF transmembrane regulator (CFTR) gene) while minimising the detection of unaffected carriers of CF. However, when carrier results are identified, these too need to be communicated to families. Similar to SCD, this occurs in a range of ways by different people, including general practitioners, health visitors, specialist nurses and genetic counsellors via telephone or home visits.

> CF carriers, we've got two specialist nurses within our screening lab and they actually go and do a home visit with the families and give them the CF carrier result in person Site 13

Feedback to the NBS laboratories regarding whether the parents had been informed of their baby's SCD or CF carrier status and by whom was also of concern to some laboratories.

### Resource use and responsibilities

NBS cards are used to collect blood from infants and information from their parents when the NBS heel prick sample is taken when the child is approximately 5 days of age. Procedures for processing NBS cards for the nine conditions currently included in the NBS Programme were considered to be transparent and efficient; clinicians within the NBS laboratories were able to discuss these in detail and seemed satisfied with how the laboratory guidelines were operationalised.

However, operationalising the communication aspect of the NBS pathways had clear implications in terms of resources; communicating positive newborn screening results for the screened conditions could be time consuming for a range of reasons. These included, not being able to contact the appropriate person or needing to wait for the appropriate busy clinician to return a telephone call.

> Sometimes…you'd speak to a secretary and you'd be waiting for a doctor to get back to you. So, it can take a few hours from when you'd started to try and process that, depending on how quickly get back to you. Site 12

> …it does sometimes feel like a bit of a battle trying to get hold of someone. Site 3

Again, this was also seen as being condition specific.

> Once I know about a positive result (for CF or one of the metabolic conditions), ten, fifteen minutes….for CHT, I might be chasing around for an hour. Site 10

Due to the time-consuming nature of referring positive NBS results to the relevant professional or team, specialist screening professionals were highly valued by laboratory staff due to their knowledge of the conditions but also their ability to provide a link between the laboratory and the clinical teams. However, this service was not universally available.

> We don't have anything luxurious like a screening specialist nurse or anything like that, which I think probably would be extremely useful, but we don't. Site 2

Once the screening result has been received by the relevant member of the clinical team, a large amount of time is dedicated to organising and preparing for the family to be seen by the clinical team for the first time. The responsibility for this is shared and might be undertaken by the laboratory team, the consultant's secretary and/or the consultant or clinical nurse specialist for the specific condition. This was often centre-specific and depends on local arrangements and resources.

Although quite unusual, in some areas, the NBS laboratory would be responsible for arranging the appointment and sometimes the diagnostic tests. In others, this would be the responsibility of the specialist screening coordinator, screening nurse or screening health visitor. In other centres, this would be the responsibility of members of the relevant clinical team such as the consultant or specialist nurse or midwife.

### Communication pathways

Communication pathways for the 13 newborn screening laboratories in England for positive NBS results were developed from the interview responses. These described how positive NBS results were communicated from the NBS laboratory to parents via the clinical team and highlighted when variations occurred (online supplemental file 1).

### DISCUSSION

The findings of this study indicate that NBS laboratory staff are acutely aware of the significance of a positive NBS result and the potential impact this could have on the family, even though in the majority of cases, they do not have direct contact with the family. Despite this, some challenges existed when communicating results from laboratories to relevant clinicians. These could involve time-consuming processes being employed to ensure the NBS 'communication loop' from the laboratory to clinical teams to families and back to the laboratory was closed. Nevertheless, laboratory staff in the present study were willing to invest the additional time and effort needed to ensure the performance thresholds of the NBS

Programme were met in particular in relation to referral of screen positive samples, and timely entry into clinical care for all screen positive babies referred to specialist services.[25]

The communication pathway for positive NBS results starts in the laboratory via relevant clinical teams and ends with the family of the affected child. Many studies have explored communication of positive NBS results to families[11–16] but none have explored communication of NBS results between the laboratory and clinical teams involved in this process.

Although templates for the communication of positive NBS results exist, most laboratories had developed their own proformas, designed to meet local needs more explicitly. Although many laboratories stated these were based on the standard national templates, these had been adapted for a variety of reasons including: attempting to make the formatting of the proforma compatible with existing computer systems and data generated during processing of the NBS result; feedback from clinical teams regarding the content of information that would be useful when receiving the NBS result; and the addition of information to assist laboratories to obtain information from clinical teams about when the baby had been seen so this information could be uploaded to the Child Health Information Service (CHIS) (clinical care records for children which contains information about a child's public health interventions, such as screening, immunisations and outcomes). However, this meant that for clinical teams who received positive NBS results from more than one laboratory, the information and format used varied. It is known that variations exist both nationally and internationally in terms of the approaches used to communicate positive NBS results to families[11–16] but this would also seem to extend to the approaches used to communicate positive NBS results between clinicians. In addition, laboratory staff could spend considerable time trying to locate and make a referral to the correct clinician. Finally, obtaining the necessary feedback from clinical teams to enable the laboratories to complete their reporting processes could also be time-consuming and challenging. Although this is unlikely to influence communication with the family, it is important to ensure information relating to every child's NBS journey is documented in a timely fashion and is available to relevant professionals involved in the child's care. Interestingly, providing this feedback was not seen as being an issue by clinical staff, which suggests that they may not be aware of the information that is needed and could explain why in some cases this is not fed back in a timely manner.

The communication pathways identified some key similarities and differences between the different NBS laboratories. Key similarities included: contacting the relevant clinical team via telephone or in person (depending on physical proximity to the NBS laboratory) to alert them of a potential positive NBS positive result prior to sending the formal proforma, normally via secure email; requesting feedback from clinical teams

regarding when the baby had been seen and the outcome (although this could either be in the form of a locally generated feedback form or a request for a copy of the clinic letter generated after the initial consultation with the child and family); and automatic upload of screening outcomes to the CHIS. Differences included: who in the clinical team was contacted by the NBS laboratory. This included the Consultant, Registrar, the Clinical Nurse Specialist, Specialist Health Visitors/Midwives, Genetic Counsellors and Screening/Pathway Co-ordinators. In some instances, this was condition specific but often this was determined by local arrangements and availability of resources. The person responsible for arranging the logistics of the initial appointment with the family also varied from the Consultant to more commonly the Clinical Nurse Specialist to infrequently a member of laboratory staff or the Consultant's secretary. This variability reflects those observed when positive NBS results are communicated to families.[11–16] Availability of specialist screening nurses who could act to bridge these processes by receiving the screening result, arrange the follow-up required and deliver the positive NBS result to the family, were highly valued by both laboratory staff and members of clinical teams and viewed as an example of good practice. Communication pathways for carrier results, the most common outcome for NBS screening particularly for SCD, also varied significantly by condition and locality.

Communicating positive NBS results for CHT seemed to be particularly problematic perhaps because communication is not always via a specialist clinical team due to CHT being viewed as manageable by general paediatricians. Different models of care are in operation throughout the country and how these operate seemed to be influenced by local arrangements and resources but also to some extent historical influences. This often led to difficulties from the laboratory perspective: who the correct person was to refer a baby with suspected CHT to; a lack of confidence on occasions once the referral had been made; and concern that the child may not be followed up according to national guidelines. However, performance data from 2017 to 2018[26 27] indicates that 92.7% of babies with a CHT positive screening result had a clinical referral initiated within 3 working days of sample receipt by the NBS laboratory, this compared with 100% of babies with MSUD, GA1, IVA, MCADD and 99.1% of babies with PKU. In addition, 93% of children with CHT entered clinical care in a timely manner, this was higher than for those babies with MSUD (50.0%), PKU 61.1%), HCU (66.7%), CF (66.8%) and MCADD (76.2%). Therefore, while these performance data indicate that difficulties communicating the positive NBS result for CHT may have hindered timely referral to relevant clinical teams, this did not delay initiation of clinical care. Other conditions have dedicated clinical teams that focus solely on that condition (CF) or range of conditions (SCD and the metabolic conditions) whereas babies with CHT are seen by endocrine teams who manage a range of other unrelated conditions in addition to CHT.

As such. the performance data[26 27] indicates that referrals for these conditions were made more quickly although this was not reflected in the time taken to enter clinical care. However, this could also reflect the differing complexities of these conditions in terms of treatment. Another factor that was considered to improve the referral process, particularly for the IMDs, was the close working relationships both physically and personally between laboratory staff and clinical teams were seen to enhance the referral process. However, this could also pose a potential risk if, for instance, relationships deteriorate or people move jobs. This was not evident in the present study but is perhaps something that needs careful consideration to avoid an over-reliance on relationships rather than robust systems.

### Strengths and limitations

The current study has numerous strengths. This is the first known study that has explored communication pathways for positive NBS results from the laboratory to clinical teams. Participants represented the 13 NBS laboratories in England involved in managing the nine conditions currently included in the NBS programme, increasing the transferability of the findings. In addition, the study design, data collection and analysis were influenced by members of the PPI advisory group and relevant charities. In terms of limitations, the duration of the interviews ranged quite widely and the richness of the data collected was limited in the shorter interviews; in most instances, this was considered to reflect the limited experiences of some of the staff who were interviewed. The researchers are experienced in this field which may have biassed data collection and analysis.

### Recommendations for practice

The findings of this study suggest a range of recommended routes, with some key requirements for communication of positive NBS results from laboratories to clinicians, allowing for local variation and complexity, would be beneficial. This would ensure information sharing among the range of professionals involved nationally is optimised following a positive NBS result. This would include information required by laboratories to complete their processes, which would reduce demand on resources currently used to ensure this information is collected. Further research is needed to explore communication pathways for positive NBS results for CHT to ensure the process is streamlined and clear communication pathways are in place. While it is acknowledged that a single approach for processing CHT results may not be appropriate or necessary, transparency and complete information such as named contact individuals may help to ensure the process is less labour intensive, particularly from a laboratory perspective. In addition, to reduce variability in communication practices, the inclusion of specialist screening nurses as part of the NBS laboratory team was viewed as an example of good practice.

### CONCLUSION

The importance of ensuring timely and effective communication of positive NBS results from laboratories to families via relevant clinicians was seen as a priority for all involved. Variation exists in terms of proformas and processes used to report positive NBS results to clinical teams, the approaches used and the people involved. This was often determined by local arrangements and resources. These different approaches could be time-consuming and challenging. Despite these challenges, ensuring quality and consistency was a clear priority for all NBS laboratories.

**Author affiliations**

[1]Centre for Maternal and Child Health Research, City, University of London, London, UK

[2]Faculty of Sports, Health and Applied Science, St Mary's University Twickenham, Twickenham, UK

[3]Biochemical Sciences, Viapath, Guys & St Thomas' NHS Foundation Trust, GKT School of Medical Education, King's College London, London, UK

[4]Women's and Children's Health, University of Liverpool, Liverpool, UK

[5]Centre for Arts, Memory and Communities, Coventry University, Coventry, UK

[6]School of Clinical Medicine, University of Cambridge, Cambridge, UK

[7]Division of Psychology and Mental Health, The University of Manchester, Manchester, UK

[8]Paediatric Psychology and Play Services, Great Ormond Street Hospital for Children NHS FoundationTrust, London, UK

[9]Florence Nightingale Faculty of Nursing, Midwifery & Palliative Care, King's College London, London, UK

**Acknowledgements** The authors would like to thank the Newborn Screening Laboratory Directors in England for agreeing to act as local principal investigators for this study. The authors would also like to thank all the parents in the Public and Patient Involvement Advisory Group for this study for their invaluable input.

**Contributors** JC made substantial contributions to the conception and design of the work. She acquired and interpreted the data for the work. She was involved in drafting the work, approved the final version to be published and agrees to be accountable for all aspects of the work in ensuring that questions related to the accuracy or integrity of any part of the work are appropriately investigated and resolved. AS and SM made substantial contributions to the conception and design of the work. They were involved in drafting the work, approved the final version to be published and agree to be accountable for all aspects of the work in ensuring that questions related to the accuracy or integrity of any part of the work are appropriately investigated and resolved. EO, PH and HC were involved interpreting the data for the work and revising the work critically for important intellectual content, approved the final version to be published and agree to be accountable for all aspects of the work in ensuring that questions related to the accuracy or integrity of any part of the work are appropriately investigated and resolved. LM, FU, MB, RSC and KS made substantial contributions to the conception and design of the work. They were involved in drafting the work, approved the final version to be published and agree to be accountable for all aspects of the work in ensuring that questions related to the accuracy or integrity of any part of the work are appropriately investigated and resolved.

**Funding** This study is funded by the National Institute for Health Research (NIHR) (Health Services and Delivery Research (project reference 16/52/25)). The views expressed are those of the authors and not necessarily those of the NIHR or the Department of Health and Social Care.

**Competing interests** None declared.

**Patient consent for publication** Not required.

**Ethics approval** This study was approved by the London Stanmore ethics committee, reference 231291.

**Provenance and peer review** Not commissioned; externally peer reviewed.

**Data availability statement** Data are available upon reasonable request from the corresponding author subject to restrictions to preserve anonymity and personal privacy (JC). These data are not publicly available as they contain information that could compromise research participant privacy/consent. Data will be available beginning 1 year and ending 5 years after publication to researchers who propose a methodologically sound proposal. Proposals should be directed to j.chudleigh@city.ac.uk. To gain access, data requesters will need to sign a data access agreement.

**ORCID iDs**
Jane Chudleigh http://orcid.org/0000-0002-7334-8708
Stephen Morris http://orcid.org/0000-0002-5828-3563

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
