## [Reviewer comments · BMJ Open]

ARTICLE DETAILS

TITLE (PROVISIONAL)	Processing of positive newborn screening results: a qualitative exploration of current practice in England
AUTHORS	Chudleigh, Jane; Chinnery, Holly; Holder, Pru; Carling, Rachel; Southern, Kevin; Olander, Ellinor; Moody, Louise; Morris, Stephen; Ulph, Fiona; Bryon, Mandy; Simpson, Alan

VERSION 1 – REVIEW

REVIEWER	Tara Tancred Liverpool School of Tropical Medicine, UK
REVIEW RETURNED	07-Oct-2020

GENERAL COMMENTS	This was a really nice paper. The tables in the introduction are very helpful, particularly for readers who may be coming from outside of the UK context to understand the processes here. Results were really nice. I just have a few very minor comments, which are at your discretion: Abstract - Comma after “sickle cell disease” should be removed. Introduction - To fully appreciate the need for the study, it would be helpful to preface this a bit more in the introduction. Why study these communication pathways? Are there known gaps or problems associated with this? Are there specific implications for clinical practice that inspired the study? Methods - I realise this is also the same as your other study around communicating to parents, but it might be helpful, for this particular study and set of results, to have the inclusion criteria reflect communication between clinicians and labs. - The name “process maps” as referred to on page 12 is a bit unclear. Maybe refer to these as “communication pathways” as you do on page 19? (Note: all of these maps are not formatted correctly, which I imagine was an issue once they were uploaded, but all of the arrows and boxes are out of alignment) Discussion - On page 14, the results indicate that 10/13 labs created their own templates for referring positives. I found this quite interesting and was hoping to see this discussed a bit more, particularly in terms of why these teams did this. I think there may be important implications around the inadequacy of existing guidance/suggestions for how this might be improved. - Further, some reflection around the key similarities and differences
--

	in the communication pathways (perhaps linking to gaps in some, again with clear implications for practice) might be useful. Especially given that these labs come from all over the UK, which I imagine may vary in capacity.
--	--

REVIEWER	Beth K Potter University of Ottawa, Canada
REVIEW RETURNED	09-Oct-2020

GENERAL COMMENTS	This paper addresses an important topic, the communication practices and pathways related to positive newborn screening (NBS) results in the UK. The results have implications for the program in the UK, including for the 10,000 children who receive positive NBS results and their families; and for NBS programs in other jurisdictions. This study was very large (71 participants) for a qualitative inquiry and nicely balances breadth and depth of insight. Introduction: 1. A minor point in the second paragraph of the introduction – the authors note that positive results include those for babies with the screened disorders and who are unaffected gene carriers (for SCD and CF). A group missing from this sentence are babies who screen positive but for whom screened disease is ruled out (false positive results). Methods: 2. Regarding patient and public involvement, it is impressive that an advisory group of 8 parents was involved in the study. The authors state that, “Their suggestions were incorporated into the study design, the data collection tools and the data analysis and presentation.” This would be strengthened if the authors can give some examples of the suggestions from the PPI advisory group that were incorporated into the study. I was a little surprised in the findings that interviewees (particularly clinical providers) did not discuss communication with families of screened children and barriers and facilitators to prompt contact and provision of diagnostic care, although I understand that that has been more well-researched in the published literature whereas the laboratory/clinical service communication loops have not been investigated in previous studies. 3. The authors note that the semi-structured interviews included a number of closed and open-ended questions. Was there an interview guide? If so, please provide as an appendix. 4. I am not used to seeing measures of inter-coder reliability in qualitative research but accept that the authors perhaps used a content analysis approach that is more quantitative in nature. Having said that, if inter-coder reliability is reported, Kappa should be reported rather than percent agreement. Results and Discussion: 5. There was a very wide range in the length of the interviews (~10 minutes to ~55 minutes). I wonder if the median length rather than the mean length of interview would be the more appropriate measure to report in Table 3. This variation also suggests variability in the richness of the interview conversations. I think commenting on
---

	this would be helpful in the Discussion. 6. Some of the themes that the authors describe seem more like topics covered in the interview, rather than inductive themes that were derived from the data (e.g., “referral from the laboratory to clinical teams”; and “feedback from clinical teams to NBS laboratories” do not read as themes in the typical sense of insights that are derived from qualitative analysis). This needs to be clarified. 7. In the results for the themes around communication from laboratories to clinical services and vice versa, did communication back to the laboratory and program about final resolution as true or false positive (and not just the screening result and responsibility for follow-up) arise? I was curious about whether the communication barriers described in particular for CHT impacted the program’s ability to evaluate the clinical validity of screening (particularly false positives vs true positives) and birth prevalence of the screened disorders. 8. The communication barriers experienced by the laboratory teams in reaching clinical services were clear and well-described. However, I did not get a sense of whether this was believed to cause delays in achieving prompt diagnosis for children with the screened conditions or whether there was concern about missing cases due to communication problems (except a mention of some worry with respect to communication of carrier results). Did this issue arise in the interviews? The importance of personal relationships and knowing one another within the communication network was interesting but also concerning as it seems to place responsibility on this aspect rather than robust systems. 9. The variability in communication processes, for CHT in particular, was striking. Is there known to be regional differences in time from screening sample to diagnosis in the UK (or in related metrics such as age at diagnosis or age at treatment initiation for screen-identified CHT)? If not, is this a potentially important implication of the disparate communication pathways that requires further study?
--	--

VERSION 1 – AUTHOR RESPONSE

Reviewer: 1

Reviewer Name: Tara Tancred

Institution and Country: Liverpool School of Tropical Medicine, UK

Please state any competing interests or state ‘None declared’: None declared

This was a really nice paper. The tables in the introduction are very helpful, particularly for readers who may be coming from outside of the UK context to understand the processes here. Results were really nice. I just have a few very minor comments, which are at your discretion:

Thank you for your kind comments

Abstract

- Comma after “sickle cell disease” should be removed.

This has been removed

Introduction

- To fully appreciate the need for the study, it would be helpful to preface this a bit more in the introduction. Why study these communication pathways? Are there known gaps or problems associated with this? Are there specific implications for clinical practice that inspired the study? A more in-depth review of relevant literature has been included (Page 5)

Methods

- I realise this is also the same as your other study around communicating to parents, but it might be helpful, for this particular study and set of results, to have the inclusion criteria reflect communication between clinicians and labs.

This has been amended (page 9)

- The name “process maps” as referred to on page 12 is a bit unclear. Maybe refer to these as “communication pathways” as you do on page 19? (Note: all of these maps are not formatted correctly, which I imagine was an issue once they were uploaded, but all of the arrows and boxes are out of alignment)

Apologies, this has been amended and the pathways have been ‘grouped’ to avoid the formatting issues.

Discussion

- On page 14, the results indicate that 10/13 labs created their own templates for referring positives. I found this quite interesting and was hoping to see this discussed a bit more, particularly in terms of why these teams did this. I think there may be important implications around the inadequacy of existing guidance/suggestions for how this might be improved.

Further detail has been added regarding what prompted laboratories to adapt the national template (Page 20)

- Further, some reflection around the key similarities and differences in the communication pathways (perhaps linking to gaps in some, again with clear implications for practice) might be useful. Especially given that these labs come from all over the UK, which I imagine may vary in capacity.

This has been added to the discussion section (Page 20)

Reviewer: 2

Reviewer Name: Beth K Potter

Institution and Country: University of Ottawa, Canada

Please state any competing interests or state ‘None declared’: None declared

This paper addresses an important topic, the communication practices and pathways related to positive newborn screening (NBS) results in the UK. The results have implications for the program in the UK, including for the 10,000 children who receive positive NBS results and their families; and for NBS programs in other jurisdictions. This study was very large (71 participants) for a qualitative inquiry and nicely balances breadth and depth of insight.

Thank you for your kind comments

Introduction:

1. A minor point in the second paragraph of the introduction – the authors note that positive results include those for babies with the screened disorders and who are unaffected gene carriers (for SCD and CF). A group missing from this sentence are babies who screen positive but for whom screened disease is ruled out (false positive results).

Thank you, this has been added (Page 4).

Methods:

2. Regarding patient and public involvement, it is impressive that an advisory group of 8 parents was

involved in the study. The authors state that, “Their suggestions were incorporated into the study design, the data collection tools and the data analysis and presentation.” This would be strengthened if the authors can give some examples of the suggestions from the PPI advisory group that were incorporated into the study.

Examples of the helpful suggestions our PPI group provided has been added (Page 9)

I was a little surprised in the findings that interviewees (particularly clinical providers) did not discuss communication with families of screened children and barriers and facilitators to prompt contact and provision of diagnostic care, although I understand that that has been more well-researched in the published literature whereas the laboratory/clinical service communication loops have not been investigated in previous studies.

These findings have been published separately here:

<https://bmjopen.bmj.com/content/bmjopen/10/10/e037081.full.pdf>. This has been added on Page 10

3. The authors note that the semi-structured interviews included a number of closed and open-ended questions. Was there an interview guide? If so, please provide as an appendix.

This has been provided as Supplementary File A

4. I am not used to seeing measures of inter-coder reliability in qualitative research but accept that the authors perhaps used a content analysis approach that is more quantitative in nature. Having said that, if inter-coder reliability is reported, Kappa should be reported rather than percent agreement.

This has been removed to avoid confusion

Results and Discussion:

5. There was a very wide range in the length of the interviews (~10 minutes to ~55 minutes). I wonder if the median length rather than the mean length of interview would be the more appropriate measure to report in Table 3. This variation also suggests variability in the richness of the interview conversations. I think commenting on this would be helpful in the Discussion.

This has been changed so the median has been reported for both the length of service and the duration of the interview as both had quite large ranges. In the discussion (Page 22), this variability has been discussed as part of the limitations of the study.

6. Some of the themes that the authors describe seem more like topics covered in the interview, rather than inductive themes that were derived from the data (e.g., “referral from the laboratory to clinical teams”; and “feedback from clinical teams to NBS laboratories” do not read as themes in the typical sense of insights that are derived from qualitative analysis). This needs to be clarified.

The subtitles of the themes have been altered to better reflect the content of the themes.

7. In the results for the themes around communication from laboratories to clinical services and vice versa, did communication back to the laboratory and program about final resolution as true or false positive (and not just the screening result and responsibility for follow-up) arise? I was curious about whether the communication barriers described in particular for CHT impacted the program’s ability to evaluate the clinical validity of screening (particularly false positives vs true positives) and birth prevalence of the screened disorders.

Even though communication difficulties were apparent and more evident for some conditions, particularly CHT, laboratory staff were acutely aware of the importance of obtaining this feedback to ensure the screening process was complete and therefore, even though it meant that they often had to invest additional time, they would ensure that feedback was received so the final outcome (true positive, false positive etc) could be recorded accurately. A sentence reflecting this has been added to the results (Page 15).

8. The communication barriers experienced by the laboratory teams in reaching clinical services were

clear and well-described. However, I did not get a sense of whether this was believed to cause delays in achieving prompt diagnosis for children with the screened conditions or whether there was concern about missing cases due to communication problems (except a mention of some worry with respect to communication of carrier results). Did this issue arise in the interviews?

A sentence about this has been added to the results as above (Page 15). Laboratory staff were acutely aware of the importance of ensuring the results were followed up in a timely manner and although it was sometime challenging to get hold of the 'right' person to refer the NBS results to, laboratory staff were willing to invest whatever time was needed into this process to ensure it was done. In addition, data regarding referral time has been added to Page 21.

The importance of personal relationships and knowing one another within the communication network was interesting but also concerning as it seems to place responsibility on this aspect rather than robust systems.

Thank you for this helpful comment. Data we gathered suggested that laboratory staff and clinicians favoured this approach and actually found it strengthened the systems that were already in place. However, as you say above, this could pose a risk if relationships deteriorate or staff leave etc. We have added this to Page 22.

9. The variability in communication processes, for CHT in particular, was striking. Is there known to be regional differences in time from screening sample to diagnosis in the UK (or in related metrics such as age at diagnosis or age at treatment initiation for screen-identified CHT)? If not, is this a potentially important implication of the disparate communication pathways that requires further study?

Performance data have been added to the discussion on Page 21.

VERSION 2 – REVIEW

REVIEWER	Tara Tancred LSTM, UK
REVIEW RETURNED	20-Nov-2020

GENERAL COMMENTS	Many thanks for this revised manuscript. This all reads very clearly, and I have no further comments.
---

REVIEWER	Beth K Potter University of Ottawa Canada
REVIEW RETURNED	11-Nov-2020

GENERAL COMMENTS	I appreciate the authors' thoughtful attention to the reviewer comments and have no further suggestions.
--